# Effects of an Invasive Mud Crab on a Macroalgae-Dominated Habitat of the Baltic Sea under Different Temperature Regimes

Imtiyaz B. Beleem , Jonne Kotta and Francisco R. Barboza *

Estonian Marine Institute, University of Tartu, Mäealuse 14, 12618 Tallinn, Estonia
* Correspondence: francisco.barboza@ut.ee

**Abstract:** The risks imposed by biological invasions on marine ecosystems are increasing worldwide. The mud crab *Rhithropanopeus harrisii* has recently expanded its distribution to the northeastern Baltic Sea, where low predatory pressures and the absence of functionally similar competitors favored the establishment of the species. Few studies have addressed the effects of the mud crab on Baltic benthic communities and habitats. Even fewer have looked at the consequences of the invader on habitats dominated by *Fucus vesiculosus*, the main habitat-forming macrophyte in the Baltic Sea. The present study experimentally analyzed, under laboratory conditions, the effects of *R. harrisii* on Baltic *F. vesiculosus* habitats and associated communities under different temperatures simulating summer and winter regimes. Our results show that the effects of the mud crab are modulated by temperature, being more pronounced under summer conditions when the metabolic demands and food intake requirements are higher. The experiment provided new insights into the capacity of *R. harrisii* to disrupt recruitment in native snail populations, jeopardizing the persistence of healthy populations of key grazers in *F. vesiculosus* habitats. Moreover, our results conclusively demonstrated the capacity of the invader to decimate native blue mussel populations. The impacts on functionally relevant invertebrates can have far-reaching ecological consequences, altering the food web and disrupting entire coastal ecosystems in the Baltic Sea.

**Keywords:** invasive species; *Rhithropanopeus harrisii*; *Fucus vesiculosus*; community effects; environmental effects; temperature

## 1. Introduction

Globalization has dramatically increased the spread of species beyond their native ranges, reshaping the boundaries of species distributions and exposing native floras and faunas to new threats to their ecological performance and persistence worldwide [1,2]. The unsustainable development and intensification of human activities, and the movement of people and goods at a pace never seen in human history, have severely compromised the capacity of ecosystems to absorb and adapt to change and increased their vulnerability to the arrival and establishment of alien species [1,3–6]. The annual rate of first records of alien species has increased since the 1950s for most taxonomic groups, with no or only marginal signs of slowing down [7–9]. Recent global estimations show that a fourth of the new first records are of species not previously described as alien elsewhere and that up to 16% of Earth species have the potential to become alien [10]. The risk of biological invasions is increasing worldwide and the efforts to prevent them seem insufficient and ineffective [5,10]. In this context, research aiming at describing and understanding the actual and potential effects of alien species has become essential to generate meaningful predictions of expected changes in terrestrial and marine ecosystems in the face of the worsening and apparently inevitable threats imposed by biological invasions.

Benthic decapods play a critical ecological role in the marine ecosystems where they occur. Often described as generalists, benthic decapods feed at different trophic levels and display diverse feeding strategies ranging from detritivorous and scavenging behavior to

grazing on algae and plants and predation on other benthic invertebrates [11]. Through their feeding activity, decapods can exert a strong control on different community components directly via predator–prey interactions or indirectly due to the propagation of effects through trophic cascades across benthic food webs, affecting major ecosystem functions such as primary production and nutrient cycling [11–13]. Non-trophic interactions also mediate the effects of decapods, as these organisms compete with other species for resources and can alter the characteristics of their habitats. Crabs and lobsters, for example, can aggressively interfere with access to food by other species and displace competitors through agonistic displays or direct engagement in combat [11]. Multiple decapod species exhibit digging behavior, producing burrows in soft-bottom habitats to avoid predators, escape environmental stress, or as housing during mating and molting [14,15]. Their burrowing promotes sediment turnover and intensifies the water–sediment exchange of gases, organic matter, and nutrients, affecting not only the physical-chemical conditions of the habitat but also the composition of resident micro- and macro-communities (e.g., [15–19]). When present in vegetated habitats, such as seagrass meadows or macroalgal beds, decapods can shred blades and dislodge whole plants during their feeding and burrowing activities, thereby modifying the structure of these habitats (e.g., [20]). Thus, the arrival and establishment of decapods can trigger structural and functional shifts in invaded ecosystems as observational and experimental evidence has shown for different species across habitats and regions (e.g., [21–25]).

Decapods are one of the most successful spreading invaders worldwide [26,27]. A combination of physiological, behavioral, life history, and ecological traits explain their successful spread and establishment. Among these traits, the most prominent are: (i) their ability to live in and plastically respond to a wide diversity of habitats and a range of environmental conditions and changes; (ii) their capacity to feed on a variety of items of plant and animal origin; (iii) the fact that a large number of decapod species are r-strategists, showing multiple and highly productive reproductive events that result in the release of pelagic stages that disperse in the water column; and (iv) their aggressivity and territorialism which transform them into tough predators, preys, and competitors (Rato et al. [27] and citations therein).

Several brachyuran decapods of Asian and American origin are successfully spreading and establishing across European Seas (e.g., [28–31]). Among them, the mud crab *Rhithropanopeus harrisii* (Gould, 1841) has successfully colonized multiple locations across the North Sea and the Baltic Sea [28,32]. Originally from North America, where its distribution extends from the west coast of Mexico and the United States to the southwest coast of Canada, the species was first observed in Europe in 1874 in the Netherlands. The first record of the species in the Baltic Sea is from Germany in 1936, from where it expanded to Poland (first record in 1951), and Lithuania (first record in 2000) (Kotta and Ojaveer [28], Fowler et al. [32] and citations therein). Only in the last decade, the species has expanded its Baltic distribution further north, invading the southwest coast of Finland (first record in 2009, [32]), Pärnu Bay, and close-by areas of the Gulf of Riga in Estonia (first record in 2011) [22,28]. The wide thermal tolerance and resistance to the extremely low salinity conditions of adults and early stages allowed the species to survive the cold winters and successfully reproduce in the diluted waters of the Baltic Sea [33,34]. In this region, *R. harrisii* has been shown to live both in macroalgae-dominated and soft-bottom habitats [35,36], where it actively feeds on bivalves, gastropods, polychaetes, other crustaceans, macroalgae, and detritus (e.g., [22,37–42]).

Only a handful of studies have analyzed, under field and laboratory conditions, the effects of *R. harrisii* on vegetated and soft-sediment habitats and associated native benthic communities in the Baltic Sea (e.g., [22,39,41,43,44]). Of these studies, few have analyzed the effects on habitats dominated by *Fucus vesiculosus* [39,41], the main canopy-forming macroalga in the Baltic Sea [45]. Available information on the impact of *R. harrisii* on Baltic *F. vesiculosus* habitats comes from field experiments, which show clear signs of deterioration in aggregated diversity indices and overall changes in the structure of

invertebrate communities in treatments where the crab was present [39,41]. The results suggest that predation impacts might result in cascading effects, releasing filamentous algae from the grazing pressure imposed by native macroinvertebrates and restructuring resident macrophyte communities [41]. However, the described effects of the mud crab on *Fucus* habitats have not been confirmed experimentally under laboratory conditions and, therefore, potential effects of confounding biotic and abiotic factors have not been controlled. In addition, most studies on the consequences of the mud crab have focused on feeding-mediated effects on resident benthic communities, generally overlooking the abiotic changes that the activity of the invader might have in Baltic Sea habitats (but see Kotta et al. [22] for a field experiment showing changes in nutrient and chlorophyll *a* concentrations likely mediated by the activity of *R. harrisii*).

In this study, the ecological and environmental effects of *R. harrisii* on *F. vesiculosus* habitats and associated invertebrate communities in the northern Baltic Sea were investigated under laboratory conditions at different temperatures representing winter and summer regimes in this region. The experiment addressed the following questions: (i) does the activity of *R. harrisii* lead to changes in the abundance and biomass of functionally relevant invertebrate groups and the biomass of the habitat-forming species (i.e., *F. vesiculosus*)?; (ii) does the invader, either directly through its physiological activity or indirectly through feeding-mediated effects on *F. vesiculosus* and associated invertebrate communities, have an impact on water quality (defined by the measurement of nutrient and chlorophyll *a* concentrations in the water)? and (iii) how do the simulated seasonal regimes modulate both the ecological and abiotic effects of *R. harrisii*?

## 2. Materials and Methods

### 2.1. Collection and Acclimation of Mud Crabs

*R. harrisii* individuals were collected in July 2021 in Pärnu Bay (58°21′42.1″ N, 24°28′56.8″ E) from artificial substrates placed in the area to support the spawning of pikeperch or using fish traps. The collected crabs were placed in plastic boxes with natural seawater obtained at the collection point and immediately transported to the Kõiguste field station laboratory (58°22′22.4″ N, 22°58′55.2″ E) of the Estonian Marine Institute (University of Tartu, Estonia). The water was constantly and abundantly aerated during transportation. Upon arrival at the field station, the crabs were separated into a larger number of boxes to prevent aggressive interactions and cannibalism. The crabs were acclimated in natural seawater (6 of salinity), constantly aerated, and kept at room temperature (20 °C) before the beginning of the experiment. Over the acclimation period, the mud crabs were fed with fresh *Mytilus trossulus* mussels.

### 2.2. Collection of Fucus vesiculosus and Associated Invertebrates

The *F. vesiculosus* used in the experiment were collected from a site located near the laboratory of the Kõiguste field station (58°22′13.6″ N, 22°58′43.2″ E), two days before the start of the experiment. Collected *F. vesiculosus* individuals attached to boulders no bigger than 20 cm were transported to the field station, where they were maintained in constantly aerated natural seawater until the beginning of the experiment. All *F. vesiculosus* individuals and associated boulders were inspected and invertebrates were carefully separated to prevent their random inclusion in the experimental units. Invertebrates were collected using hand nets in the macroalgal stand or during the cleaning of *F. vesiculosus* by dipping the macroalga in freshwater for a few seconds and immediately transferring swimming invertebrates to natural seawater. The collected animals were sorted using nets and sieves to collect gastropods and amphipods—the two main groups of macroinvertebrates found in *F. vesiculosus* stands in the area. Mussels of the species *M. trossulus* were collected from the pilot mussel farm located in Tagalaht Bay (58°27′36.0″ N, 22°03′00.1″ E).

### 2.3. Experimental Design and Setup

The experiment lasted one month, from 18 November to 18 December 2021. Twenty plastic boxes (size: $40 \times 60 \times 43$ cm) filled with natural seawater and continuously aerated were used as experimental units. The bottom of all the experimental units was covered with previously washed pebbles, increasing the topographic complexity of the boxes, and providing habitat for crabs and invertebrates. Two or three *F. vesiculosus* individuals (depending on their size) with their respective boulders were gently placed in each of the experimental units. Only healthy macroalgal individuals with no major signs of grazing were included. Regarding the associated macrofauna, 20 mussels (*M. trossulus*), 20 amphipods (*Gammarus* spp.), and 20 gastropods (predominantly *Theodoxus fluviatilis*) were included in each experimental unit, resembling the natural densities of invertebrates observed in the field at the time of sampling and considering the long-term information gathered by the Estonian Marine Institute during monitoring campaigns in the area.

Ten of the experimental units received two mud crabs each, recreating the densities observed in Pärnu Bay [40]. All *R. harrisii* individuals (carapace length (CL): 11.73–14.51 mm) used in the experiment were males, preventing potential differences in observed effects driven by the sex of individuals. Animals missing limbs or with major cracks or other signs of damage in the exoskeleton were excluded. In addition, mud crabs that did not defend themselves or exhibit escape behavior during handling were also excluded. The remaining ten experimental units without crabs were used as controls to assess the effects of the invader.

To test how seasonal thermal differences modulate the effects of the invader, 10 experimental units, 5 with crabs, and 5 controls, were kept inside the laboratory at a constant temperature of approximately 20 °C (summer conditions) while the remaining 10 were exposed to natural outdoor temperatures, which fluctuated between 0 and 6 °C (winter conditions) over the duration of the experiment. To prevent differences in light conditions between experimental units kept indoors and outdoors, the same fluorescent-lamp system installed in the laboratory was used for the experimental units kept outdoors. All experimental units were exposed to a 12:12 h light–dark cycle throughout the experiment.

### 2.4. Data Collection and Analysis

Temperature and salinity, the activity and mortality of the crabs, and water quality were monitored throughout the experiment.

All *F. vesiculosus* used in the experiment were carefully dry blotted and their wet weight was determined at the beginning of the experiment. This information was used in combination with a standard relationship between wet weight and dry weight (constructed for the species with individuals collected at the moment of the experiment) to determine the dry weight of macroalgal individuals at the start of the experiment. All macroalgal biomass included in the experimental units was collected, dried, and weighed at the end of the experiment. This information was combined with the initial dry weights to calculate changes in the *F. vesiculosus* biomass over the duration of the experiment.

All mud crabs were sized (CL) at the beginning and the end of the experiment. All invertebrates that remained at the end of the experiment were collected with the help of a 0.25 mm mesh-size sieve, identified to the lowest taxonomic level possible, counted, and dry weighed.

Water samples for the analysis of nutrients and chlorophyll *a* were obtained at the end of the experiment before collecting all biological samples to prevent potential disturbances. Water samples for the analysis of nutrients were collected and immediately frozen until further analysis. Total phosphorus, phosphates, nitrogen, and nitrites plus nitrates concentrations were determined using a Skalar San++ analyzer (see a detailed list of the methods and protocols used in Kotta et al. [22]). In the case of chlorophyll *a* samples, 1 L of water was collected from each experimental unit and filtered through a Whatman GF/F filter. The filters were placed in the refrigerator overnight in 50 mL tubes with 10 mL of 96%

ethanol for extraction. The concentration of chlorophyll *a* was measured using a Biochrom Libra S32 spectrophotometer.

Changes in biological (*F. vesiculosus* biomass, abundance, and biomass of associated invertebrates) and water-quality variables (nutrient and chlorophyll *a* concentrations) due to the activity of the invader under different temperature regimes were statistically analyzed using two-way analysis of variance (ANOVA). The ANOVAs were performed using the function aov from the stats package in R version 4.1.2 [46]. The presence of crabs and temperature regime were included as explanatory variables in the models, evaluating both main and interactive effects. The Tukey's honest significant difference (Tukey HSD) test was applied for post-hoc pairwise comparisons using the package emmeans [47]. The fulfillment of assumptions and the overall adequacy of the models were decided based on a detailed analysis of the plots of residuals. The analysis of residuals was performed using the R package DHARMa [48]. When needed, response variables were logarithmically transformed.

## 3. Results

The simulated seasonal regimes modulated the predation effects of *R. harrisii* on the invertebrate communities (Figure 1, Table 1). While the feeding activity of the invader did not significantly affect the abundance and biomass of amphipods (Figure 1a,b, Table 1), the abundance and biomass of the bivalves and gastropods were significantly reduced by the invader under summer conditions when compared to those observed under winter conditions (Figure 1c–f, Tables 1 and S1). The invaders decimated the bivalves under summer conditions, with abundance and biomass declining by approximately 75% (Figure 1c,d). Even though reproduction partially compensated for the decrease in the abundance of gastropods caused by the invader in the summer treatment (as shown by the higher number of gastropods observed at the end of the experiment in the warm treatments both with and without crabs), *R. harrisii* was able to significantly disrupt gastropod recruitment, keeping the overall abundance at the level observed at the beginning of the experiment (Figure 1e). The abundance of gastropods was approximately 70% lower in the warm treatment with mud crabs than in the warm treatment without the invader (Figure 1e).

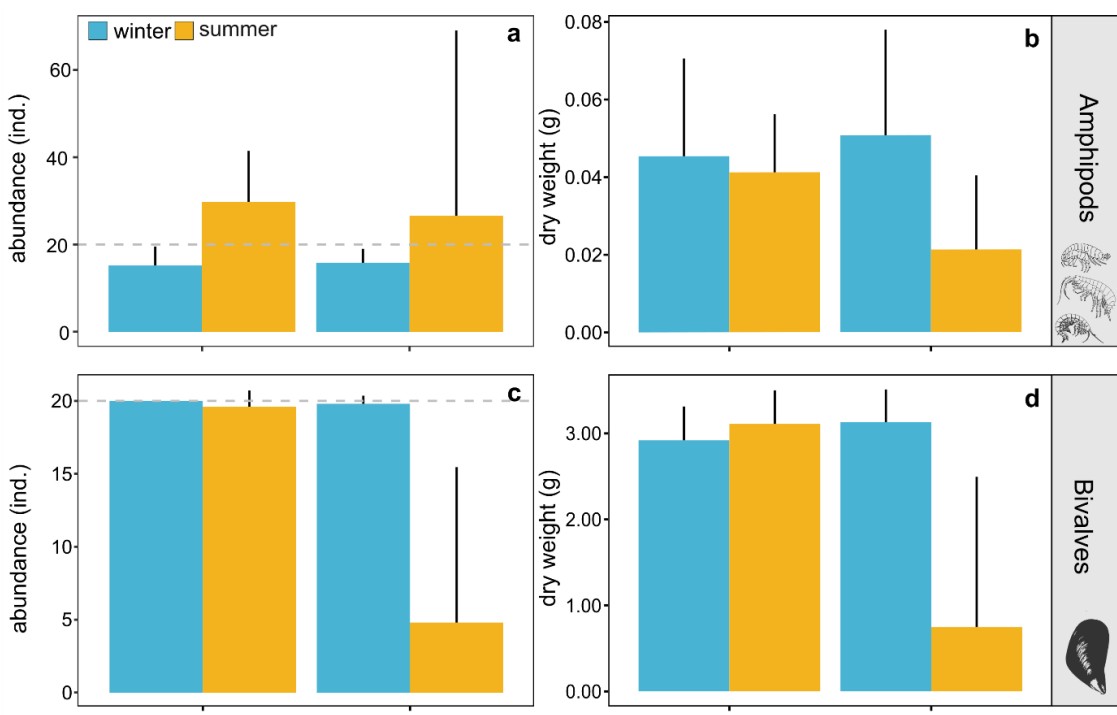

**Figure 1.** *Cont.*

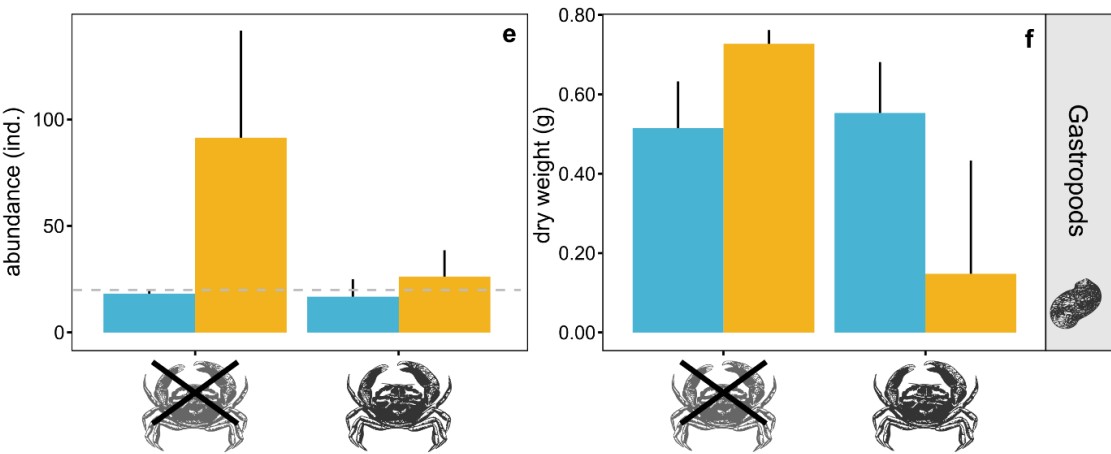

**Figure 1.** Abundance and biomass (expressed as dry weight) of amphipods (**a**,**b**), bivalves (**c**,**d**), and gastropods (**e**,**f**) observed at the end of the experiment in treatments with and without the mud crab *Rhithropanopeus harrisii* under simulated summer (yellow) and winter (blue) conditions. Bars represent mean values and whiskers confidence intervals at 95%. Horizontal dashed lines indicate the abundance of the different groups at the beginning of the experiment.

**Table 1.** Analysis of variance (ANOVA) tables for the biological response variables assessed. Model components (including both the main and interactive effects of temperature and the presence of crabs, Rh), degrees of freedom (DF), the sum of squares (SS), the mean sum of squares (MSS), the statistic (F-value), and associated significance (*p*-value) are presented. For significant effects, *p*-values are highlighted in bold. Logarithmically transformed variables are indicated with log.

| Response Variable | Model Component | DF | SS | MSS | F-Value | *p*-Value |
|---|---|---|---|---|---|---|
| log(abundance of amphipods) | temperature | 1 | 0.01 | 0.01 | 0.008 | 0.9300 |
| | Rh | 1 | 2.04 | 2.04 | 1.953 | 0.1810 |
| | temperature:Rh | 1 | 2.41 | 2.40 | 2.307 | 0.1480 |
| | residuals | 16 | 16.68 | 1.04 | | |
| log(abundance of bivalves) | temperature | 1 | 11.91 | 11.91 | 9.128 | **0.0081** |
| | Rh | 1 | 11.75 | 11.75 | 9.001 | **0.0085** |
| | temperature:Rh | 1 | 11.59 | 11.59 | 8.882 | **0.0088** |
| | residuals | 16 | 20.88 | 1.31 | | |
| log(abundance of gastropods) | temperature | 1 | 5.00 | 5.00 | 22.64 | **0.0002** |
| | Rh | 1 | 2.38 | 2.38 | 10.75 | **0.0047** |
| | temperature:Rh | 1 | 1.24 | 1.24 | 5.63 | **0.0305** |
| | residuals | 16 | 3.54 | 0.22 | | |
| log(dry weight of amphipods) | temperature | 1 | 1.36 | 1.36 | 4.063 | 0.0610 |
| | Rh | 1 | 0.78 | 0.78 | 2.333 | 0.1462 |
| | temperature:Rh | 1 | 1.14 | 1.14 | 3.399 | 0.0838 |
| | residuals | 16 | 5.35 | 0.33 | | |
| log(dry weight of bivalve) | temperature | 1 | 6.45 | 6.45 | 11.57 | **0.0036** |
| | Rh | 1 | 6.37 | 6.37 | 11.43 | **0.0038** |
| | temperature:Rh | 1 | 7.17 | 7.17 | 12.87 | **0.0025** |
| | residuals | 16 | 8.91 | 0.56 | | |

**Table 1.** *Cont.*

| Response Variable | Model Component | DF | SS | MSS | F-Value | *p*-Value |
|---|---|---|---|---|---|---|
| dry weight of gastropods | temperature | 1 | 0.05 | 0.05 | 2.53 | 0.1315 |
| | Rh | 1 | 0.37 | 0.37 | 19.93 | **0.0004** |
| | temperature:Rh | 1 | 0.48 | 0.48 | 25.97 | **0.0001** |
| | residuals | 16 | 0.29 | 0.02 | | |
| decrease in dry weight of *Fucus vesiculosus* | temperature | 1 | 1558.30 | 1558.30 | 23.04 | **0.0002** |
| | Rh | 1 | 122.20 | 122.20 | 1.81 | 0.1977 |
| | temperature:Rh | 1 | 409.90 | 409.90 | 6.06 | **0.0256** |
| | residuals | 16 | 1082.30 | 67.60 | | |

The dry weight of the macroalgae decreased in all treatments over the course of the experiment, with a greater decrease in the warm treatments and especially in the treatment without crabs (Figure 2). The decrease in biomass in the warmer treatment without crabs was on average 2–3 times higher than in the colder treatments (with and without crabs) and 1.5 times higher, but marginally significant, than in the warm treatment with mud crabs (Figure 2, Tables 1 and S1). The latter may suggest that the grazing pressure on *F. vesiculosus* is reduced as a result of the presence or feeding activity of the invader.

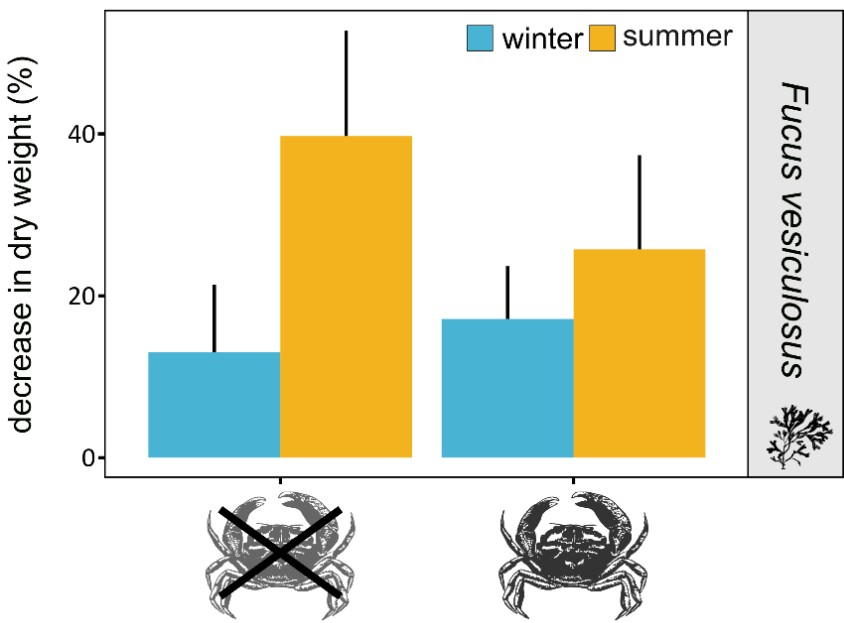

**Figure 2.** Decrease in biomass (dry weight) of *Fucus vesiculosus* over the duration of the experiment in treatments with and without the mud crab *Rhithropanopeus harrisii* under simulated summer (yellow) and winter (blue) conditions. Bars represent mean values and whiskers confidence intervals at 95%.

Although differences were, in general, not significant, on average concentrations of nitrogen-related nutrients tended to be higher in the warm treatment with crabs than in the same treatment without the invader (Figure 3a–c, Table 2). This trend was not observed for phosphorus-related nutrients (Figure 3d,e, Table 2). No significant differences were observed for chlorophyll *a* between treatments with and without mud crabs (Figure 3f, Table 2).

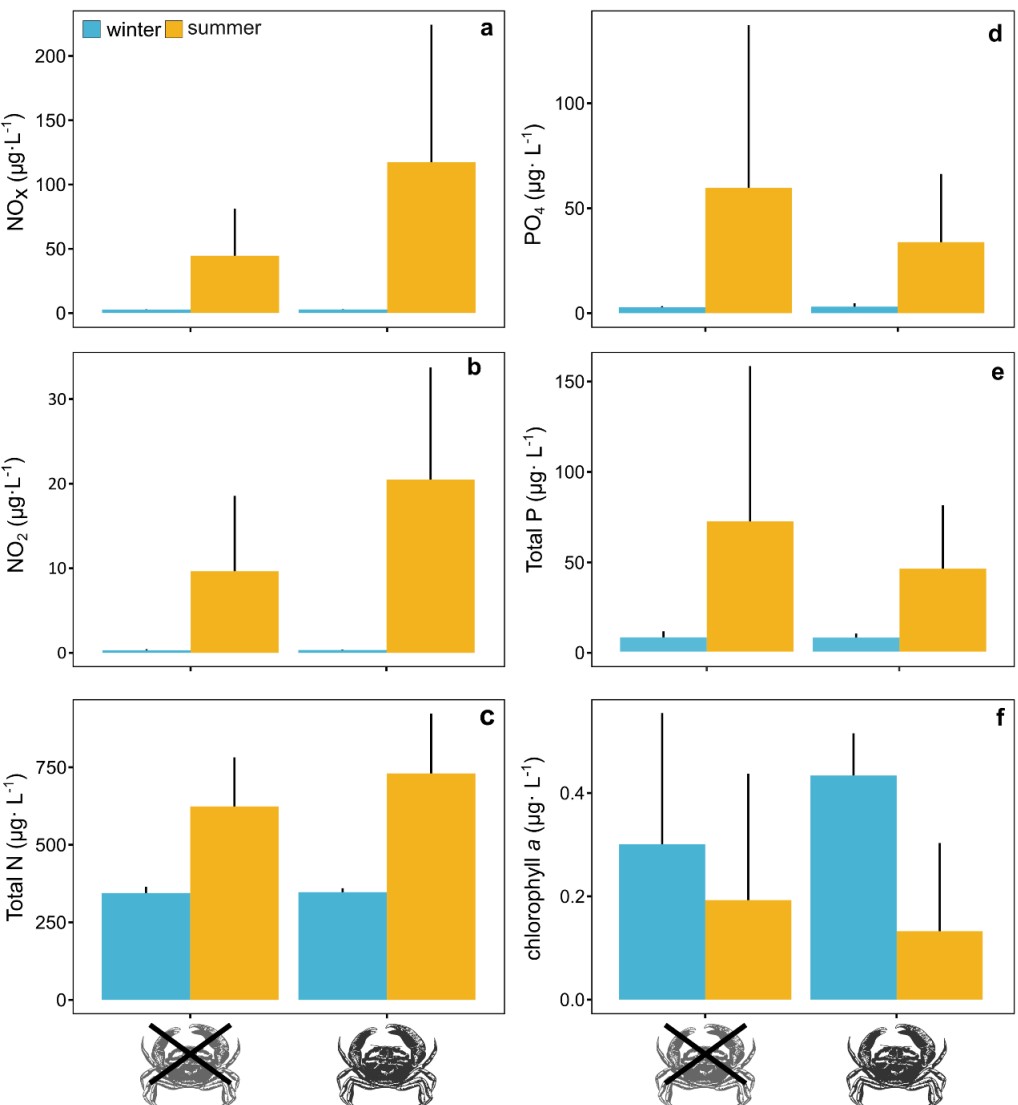

**Figure 3.** Nutrient (**a**–**e**) and chlorophyll *a* (**f**) concentrations in treatments with and without the mud crab *Rhithropanopeus harrisii* under simulated summer (yellow) and winter (blue) conditions. Bars represent mean values and whiskers confidence intervals at 95%.

**Table 2.** Analysis of variance (ANOVA) tables for the environmental response variables assessed. Model components (including both the main and interactive effects of temperature and the presence of crabs, Rh), degrees of freedom (DF), the sum of squares (SS), the mean sum of squares (MSS), the statistic (F-value), and associated significance (*p*-value) are presented. For significant effects, *p*-values were highlighted in bold. Logarithmically transformed variables are indicated with log.

| Response Variable | Model Component | DF | SS | MSS | F-Value | *p*-Value |
|---|---|---|---|---|---|---|
| log(Total N) | temperature | 1 | 2.12 | 2.12 | 92.16 | **<0.0001** |
| | Rh | 1 | 0.03 | 0.03 | 1.44 | 0.2470 |
| | temperature:Rh | 1 | 0.03 | 0.03 | 1.12 | 0.3070 |
| | residuals | 16 | 0.37 | 0.02 | | |
| log(NO$_X$) | temperature | 1 | 45.27 | 45.27 | 115.42 | **<0.0001** |
| | Rh | 1 | 1.28 | 1.28 | 3.27 | 0.0895 |
| | temperature:Rh | 1 | 1.02 | 1.02 | 2.60 | 0.1261 |
| | residuals | 16 | 6.27 | 0.39 | | |

**Table 2.** *Cont.*

| Response Variable | Model Component | DF | SS | MSS | F-Value | *p*-Value |
|---|---|---|---|---|---|---|
| log(NO$_2$) | temperature | 1 | 66.07 | 66.07 | 218.42 | **<0.0001** |
| | Rh | 1 | 1.39 | 1.39 | 4.61 | **0.0475** |
| | temperature:Rh | 1 | 0.74 | 0.74 | 2.46 | 0.1363 |
| | residuals | 16 | 4.84 | 0.30 | | |
| log(Total P) | temperature | 1 | 16.00 | 16.00 | 48.90 | **<0.0001** |
| | Rh | 1 | 0.11 | 0.11 | 0.34 | 0.5690 |
| | temperature:Rh | 1 | 0.11 | 0.11 | 0.33 | 0.5740 |
| | residuals | 16 | 5.24 | 0.33 | | |
| log(PO$_4$) | temperature | 1 | 29.10 | 29.10 | 61.97 | **<0.0001** |
| | Rh | 1 | 0.23 | 0.23 | 0.49 | 0.4940 |
| | temperature:Rh | 1 | 0.27 | 0.27 | 0.57 | 0.4630 |
| | residuals | 16 | 7.51 | 0.47 | | |
| chlorophyll *a* | temperature | 1 | 0.21 | 0.21 | 8.08 | **0.0177** |
| | Rh | 1 | 0.01 | 0.01 | 0.25 | 0.6227 |
| | temperature:Rh | 1 | 0.05 | 0.05 | 1.79 | 0.1996 |
| | residuals | 16 | 0.42 | 0.03 | | |

## 4. Discussion

Our study demonstrated that temperature plays an important role in modulating the effects of the invasive mud crab *R. harrisii* on native *F. vesiculosus* stands and associated invertebrates in the northeastern Baltic Sea. The impact of the crab was more pronounced at higher temperatures, suggesting that the presence of the invader has negative consequences for native invertebrate species, particularly under summer conditions when the metabolic demands and food intake requirements of the invader are higher [49].

The experiment showed that *R. harrisii* can exert a strong predation pressure on native snails under summer conditions. These findings are in line with the observations made in a field experiment by Jormalainen et al. [41], who showed that the increase in the density of the mud crab in *F. vesiculosus* stands over three consecutive years was correlated with the decrease in the abundance of the native snails *Hydrobia* spp. and *T. fluviatilis*. Interestingly, our results provide new insights into the capacity of *R. harrisii* to disrupt the recruitment of native gastropods in the northeastern Baltic Sea. New snail recruits were observed in all experimental units kept at 20 °C, even though the numbers were significantly lower in the units where mud crabs were included. This suggests that the invader actively feeds on juvenile gastropods and has the potential to impair the population renewal of important grazers in *F. vesiculosus* stands. Invasive crabs of different species have been shown to predate on the larval stages and juveniles of native invertebrate species, suppressing their recruitment and disrupting natural population dynamics (e.g., [50–52]). Thus, the further expansion and increase in density of *R. harrisii* in *F. vesiculosus* habitats might impose a severe risk for the persistence of healthy native populations of gastropods in the northeastern Baltic Sea.

While previous field experiments performed in the Baltic Sea have been inconclusive about the consequences of *R. harrisii* on blue mussels [39,41,44], our results clearly show that the invader has the capacity to decimate native mussel populations in this region. Evidence from soft-sediment habitats suggests that the mud crab feeds voraciously on bivalves [22], the dominant filter-feeders in coastal areas of the Baltic Sea, and responsible for the flux of matter and energy from pelagic to benthic habitats [53,54]. Through their feeding activity, filter feeders reduce phytoplankton biomass in the water column, capturing and storing nutrients in the process. Thus, the reduction in filter feeders can exacerbate the symptoms of eutrophication [22]. This highlights the importance of understanding the tipping points of interactions between *R. harrisii* and native species to effectively predict changes in the functioning of Baltic Sea ecosystems invaded by the mud crab.

Despite the direct negative effects on benthic invertebrates, the study also found trends that might suggest the positive effects of the invader on *F. vesiculosus*. The apparent lower loss of *F. vesiculosus* biomass in the warmer treatment with *R. harrisii* than in the treatment without the crab, suggests that predation pressure, or just the presence of the invader (that can trigger escape behavior of prey, [55]), might repress the activity of grazers and reduce herbivory on *F. vesiculosus*. The spreading of cascading effects triggered by the feeding activity of *R. harrisii* on native grazers in *F. vesiculosus* stands has been previously described. However, in contrast to our results, previous evidence suggests that the feeding pressure imposed by the mud crab on mid-trophic consumers might release filamentous epiphytes from grazing, allowing them to thrive, decreasing the performance of the habitat-forming species [41]. The overall low coverage of epiphytic algae observed on *F. vesiculosus* at the beginning of our experiment, which were collected in November when the productivity of ephemeral algae is already low, might explain the differences between our results and the evidence presented by previous research. Alternatively, the relatively low proliferation of epiphytes in warmer treatments over the duration of the experiment could be the result of the grazing pressure imposed by amphipods and gastropods in the treatment without crabs, and by the direct consumption of epiphytic algae by *R. harrisii* in the treatment including the invader. Future studies should analyze the relative contribution of direct and indirect processes triggered by *R. harrisii* under different seasonal—both biological and environmental—scenarios to provide a thorough mechanistic understanding and accurate predictions of the invader's impact in Baltic *F. vesiculosus* dominated ecosystems. Year-round experiments performed in mesocosm facilities able to recreate near-natural conditions (e.g., [56,57]), would provide the ideal context to evaluate carry-over effects of the invader on the structure and functioning of native ecosystems throughout the seasons.

Even if not statistically significant, the higher mean concentrations of nitrogen-related nutrients in the warm treatment with the invader than in the control suggests that the physiological and feeding activity of the mud crab might increase the nitrogen concentrations in the water column in summer. The overall absence of effects in chlorophyll *a* could be explained by the fact that the experiment was performed in a low productivity season, as clearly shown by the low concentration registered in all the experimental units by the end of the experiment. Future studies should assess the impacts of *R. harrisii* on water quality, and abiotic conditions in general, through experiments performed over longer periods of time and considering different seasonal contexts.

To conclude, the experiment showed that the invasive crab *R. harrisii* poses a serious threat to the native mussel and gastropod populations in *F. vesiculosus* dominated ecosystems of the Baltic Sea, especially in summer. The study also highlighted the importance of understanding the complex interactions between invasive species, native species, and abiotic factors, such as temperature, in predicting the impact of invasive species on ecosystems. Once established in macrophyte-dominated habitats of the Baltic Sea, *R. harrisii* could potentially have several negative ecological impacts on these low-diversity and functionally poor ecosystems. The invasive species has almost no natural predators or competitors in the northeastern Baltic Sea. As a voracious predator that feeds on a variety of prey, its introduction could have a significant impact on native populations of these organisms, which might have far-reaching ecological consequences, altering the food web and disrupting entire coastal ecosystems in the Baltic Sea.

**Supplementary Materials:** The following supporting information can be downloaded at: https://www.mdpi.com/article/10.3390/d15050644/s1, Table S1: Tukey's Honest Significant Difference (Tukey HSD) test for those biological and environmental response variables that exhibited significant main effects of the mud crab *Rhithropopeus harrisii* or interactive effects of the invader and temperature in the analysis of variance (ANOVA). The contrast between different treatments (CW: control, winter; CS: control, summer; RhW: crab, winter; RhS: crab, summer), mean estimates (estimates), standard errors (SE), degrees of freedom (DF), the statistic (t-value) and associated significance (*p*-value) are presented. Logarithmically transformed variables are indicated with log.

**Author Contributions:** Conceptualization, J.K. and F.R.B.; Methodology, F.R.B. and J.K.; Formal Analysis, F.R.B. and I.B.B.; Investigation, I.B.B., F.R.B. and J.K.; Resources, J.K.; Data Curation, F.R.B. and I.B.B.; Writing—Original Draft Preparation, F.R.B.; Writing—Review & Editing, J.K. and I.B.B.; Visualization, I.B.B. and F.R.B.; Supervision, F.R.B. and J.K.; Project Administration, J.K. and I.B.B.; Funding Acquisition, J.K. and I.B.B. All authors have read and agreed to the published version of the manuscript.

**Funding:** This research was funded by the Estonian Research Council through the Mobilitas Pluss Programme (MOBJD1044). It received financial support from the EEA grant "Climate Change Mitigation and Adaptation" call I "Ecosystem resilience increased" project "Impacts of invasive alien species and climate change on marine ecosystems in Estonia" (2014-2021.1.06.21-0040).

**Institutional Review Board Statement:** Not applicable.

**Informed Consent Statement:** Not applicable.

**Data Availability Statement:** Data will be made available upon request.

**Acknowledgments:** We express our gratitude to Teemar Püss and Vivian Tamm for technical support over the duration of the experiment, and Kristiina Nõomaa for inspiring discussions that contributed to enriching the final version of the manuscript. Special thanks to Karolin Teeveer and Trude Taevere for the processing of biological samples, Marko Rõõmusoks for the analysis of nutrient samples, and Ilmar Kotta for the processing of chlorophyll *a* samples.

**Conflicts of Interest:** The authors declare no conflict of interest.

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
