# Peer review of "Effects of an Invasive Mud Crab on a Macroalgae-Dominated Habitat of the Baltic Sea under Different Temperature Regimes"

_diversity, doi:10.3390/d15050644_

Round 1

Reviewer 1 Report

Can you locate any information on the ecological role of R. harrisii in its native habitat in North America?  How does it compare with what you found in the Baltic Sera? Your reviewer lives in a state where R. harrisii is almost entirely confined to estuarine waters or drought-stricken areas with unusually high salinity. Macroalgae are absent in our muddy coastal areas.

The introduction is too long. It should concentrate on R. harrisii and not crabs in general.

If available, it would be useful to compare the results of this study with any comparable work on R. harrisii in its native habitat.

All names of genera and species should be in italics.

Does the green crab (Carcinus maenas) occur in your area? It is known to prey on R. harrisii.

Author Response

Reviewer 1

Can you locate any information on the ecological role of R. harrisii in its native habitat in North America?  How does it compare with what you found in the Baltic Sera? Your reviewer lives in a state where R. harrisii is almost entirely confined to estuarine waters or drought-stricken areas with unusually high salinity. Macroalgae are absent in our muddy coastal areas.

R: We appreciate the interest of the Reviewer in our study and potential comparisons with studies produced in North America. Finding meaningful research on the ecological role of R. harrisii in its native range has not been easy. Most studies from the native range have focused on physiological or developmental aspects, not on the ecological role of the species. Maybe the Reviewer is aware of some publications we might have overlooked and could share them with us.

The introduction is too long. It should concentrate on R. harrisii and not crabs in general.

R: We appreciate the suggestion of the Reviewer, which certainly tries to make the paper more concrete and easier to read. However, we think the content provided in the current version of the Introduction gives a broader and more complete background to understand the nature and implications of obtained results. By focusing the Introduction only on R. harrisii, a relatively understudied crab species, we will be overlooking relevant information generated for decapods and brachyurans that provide relevant insights to understand the effects of the species of interest.

If available, it would be useful to compare the results of this study with any comparable work on R. harrisii in its native habitat.

R: We appreciate the suggestion and understand the interest of the Reviewer in a potential comparison between the results obtained in our study and those potentially observed in the native range of R. harrisii. However, we must highlight that such a comparison is not possible since studies from the native range have focused mainly on physiological and developmental aspects and not on the ecological role of the species. In the native range, R. harrisii is one of many crab species, not generating the same interest as in the northeastern Baltic Sea, where it is invasive and no other functionally similar species occur. Maybe the Reviewer is aware of some publications we might have overlooked and could share them with us.

All names of genera and species should be in italics.

R: Thank you for highlighting this issue. All species names were included in italics in the original version of the manuscript, but some formatting issues probably happened during the submission process. We have now solved this issue by manually changing the format of the species names in the document automatically generated by the editorial service.

Does the green crab (Carcinus maenas) occur in your area? It is known to prey on R. harrisii.

R: The green crab Carcinus maenas does not occur in the northeastern Baltic Sea, the area of interest for our study. The distribution of this species is limited to the more saline southwestern Baltic Sea, occurring mainly along the Baltic coasts of Denmark and Germany and the west coast of Sweden (Bonsdorff 2006). 

References

Bonsdorff, E. (2006). Zoobenthic diversity-gradients in the Baltic Sea: continuous post-glacial succession in a stressed ecosystem. Journal of Experimental Marine Biology and Ecology, 330(1), 383-391.

Reviewer 2 Report

The manuscript by Beleem, Kotta, Barboso, entitled “Effects of an invasive mud crab on a macroalgae-dominated habitat of the Baltic Sea under different temperature regimes” is a well conceived and well written account of a laboratory experiment to demonstrate the effects of predation by the introduced crab R. harrisii on a Baltic simplified assemblage mimicking a Fucus community.

The experimental set up is simple but convincing and the statistical analysis demonstrates the direct and indirect effect of the predation by the alien decapod on the food chain of an important Baltic habitat. The manuscript is worth publication because it adds new elements on the ecological impact of an alien species.

However, I suggest considering a few amendments to the text, in order to tackle some points that I have raised, especially about the following items:

1.      The species names are to be written in Italics

2.      It could be useful to explicitly state the difference between the introduction of alien species and the invasive status (Line 40) of some of them (among which R. harrisii)

3.      The biological traits (line 76) of R. harrisii, which have been described from the literature to explain the success of its invasion in the Baltic, are repeatedly attributed (lines 46, 70) to populations in the native range. This insistence is not justified, in my opinion, because these traits are likely to be displayed in the invaded range (and this might have been observed elsewhere).

4.      The two temperature regimes that have been experimented during the one-month duration of the experiment should not be defined as “extremes” (lines 15, 129). This term conveys the idea that some unusually high or low temperatures are considered, while apparently the study wisely considers temperatures normally encountered in the Baltic region where the algae and animals are taken.

5.      The data on nutrients and chlorophyll are an interesting complement to the experiment, but in my opinion should not be over-emphasized (for instance lines 281-285, 337-340 and 367-370), not only because the differences found are not significant, nor an effect on such parameters can be expected due to the simplification of the experimental setup and the short duration of the experiment.

Some additional notes, comments and suggestions are to be found in the notes added to the enclosed pdf file of the manuscript.

Author Response

Reviewer 2

The manuscript by Beleem, Kotta, Barboso, entitled “Effects of an invasive mud crab on a macroalgae-dominated habitat of the Baltic Sea under different temperature regimes” is a well conceived and well written account of a laboratory experiment to demonstrate the effects of predation by the introduced crab R. harrisii on a Baltic simplified assemblage mimicking a Fucus community.

The experimental set up is simple but convincing and the statistical analysis demonstrates the direct and indirect effect of the predation by the alien decapod on the food chain of an important Baltic habitat. The manuscript is worth publication because it adds new elements on the ecological impact of an alien species.

R: We appreciate the positive feedback and constructive suggestions.

However, I suggest considering a few amendments to the text, in order to tackle some points that I have raised, especially about the following items:

  1. The species names are to be written in Italics

R: Thank you for highlighting this issue. All species names were included in italics in the original version of the manuscript, but some formatting issues probably happened during the submission process. We have now solved this issue by manually changing the format of the species names in the document automatically generated by the editorial service.

  1. It could be useful to explicitly state the difference between the introduction of alien species and the invasive status (Line 40) of some of them (among which  harrisii)

R: We think that explicitly stating the difference between alien species and their invasive status is out of the scope of our manuscript and will only increase the complexity and length of the Introduction. As presented in the manuscript, we are not saying that every alien species is becoming invasive, just that the increase in the number of first records of alien species increases the risk of new biological invasions (i.e., the higher the number of first records, the higher the probability of invasion).

  1. The biological traits (line 76) of  harrisii,which have been described from the literature to explain the success of its invasion in the Baltic, are repeatedly attributed (lines 46, 70) to populations in the native range. This insistence is not justified, in my opinion, because these traits are likely to be displayed in the invaded range (and this might have been observed elsewhere).

R: First, we would like to clarify that the information provided in lines 46, 70 and 76 (and the associated paragraphs) does not refer to R. harrisii but to decapods in general (as was clearly stated in the manuscript). Second, considering the comment made by the Reviewer, we have decided to remove the direct references to the native ranges included in lines 46 and 70 of the original version of the manuscript.

Regarding line 76 and the overall associated paragraph, the presented list of traits provides a general overview of the main characteristics highlighted in the bibliography (particularly in the excellent review paper by Rato et al. 2021) as those underlying the success of decapods as invaders. Here no references have been made to either native or invaded ranges.

  1. The two temperature regimes that have been experimented during the one-month duration of the experiment should not be defined as “extremes” (lines 15, 129). This term conveys the idea that some unusually high or low temperatures are considered, while apparently the study wisely considers temperatures normally encountered in the Baltic region where the algae and animals are taken.

R: We agree with the Reviewer and decided to remove the term “extreme” when referring to the temperature regimes considered in the experiment.

  1. The data on nutrients and chlorophyll are an interesting complement to the experiment, but in my opinion should not be over-emphasized (for instance lines 281-285, 337-340 and 367-370), not only because the differences found are not significant, nor an effect on such parameters can be expected due to the simplification of the experimental setup and the short duration of the experiment.

R: Attending the suggestion of the Reviewer, we have now toned down the discussion presented in lines 368-379 by reducing the speculation on the potential consequences of the species on eutrophication and explicitly mentioning the need for experiments “performed over longer periods of time and considering different seasonal contexts” to obtain conclusive results on the effects of the species on water quality.

Some additional notes, comments and suggestions are to be found in the notes added to the enclosed pdf file of the manuscript.

R: Thank you so much for the detailed revision of the manuscript. The minor grammar and syntax modifications suggested by the Reviewer in the enclosed PDF file have all been considered and included in the new version of the manuscript. Most comments related to content have been answered in the previous points. Below we provide answers to some relevant aspects highlighted by the Reviewer only in the PDF file:

Line 194. So outdoor aquaria were shielded from natural light?

R: Aquaria kept outdoors were strategically placed next to the laboratory building to reduce the incidence of natural light as much as possible.

Line 198. Why not the mortality of other invertebrates?

R: The monitoring of crabs' mortality was done systematically in all experimental units to guarantee that the predation pressure was maintained throughout the experiment and across replicates. However, the mortality of other invertebrates was evaluated at the end of the experiment. Only general observations were made during the experiment since amphipods and gastropods are small and tend to be tightly associated with the habitat-former. Therefore, systematically checking the mortality of other invertebrates throughout the experiment would have required manipulating and severely disturbing the experimental units.

Line 211. This is a bit surprising, because I was convinced that only three species were introduced in the experimental aquaria

R: The identification of all individuals included in the experimental units to the lowest taxonomic level possible was performed as a routine procedure to confirm the identity of the individuals included in the experiment.

Line 268. Why percent decrease is shown instead of absolute weights, in analogy with invertebrate data?

R: We decided to present the results as precent decrease to prevent misleading conclusions since the biomass of Fucus vesiculosus included at the beginning of the experiment was similar among the experimental units but not exactly the same.